# Influence of Manure Application on the Soil Bacterial Microbiome in Integrated Crop-Livestock Farms in Maryland

**DOI:** 10.3390/microorganisms9122586

**Published:** 2021-12-15

**Authors:** Mengfei Peng, Zajeba Tabashsum, Patricia Millner, Salina Parveen, Debabrata Biswas

**Affiliations:** 1Department of Animal and Avian Sciences, University of Maryland, College Park, MD 20742, USA; murphy7@terpmail.umd.edu; 2Biological Sciences Program-Cellular and Molecular Biology, University of Maryland, College Park, MD 20742, USA; ztabashs@terpmail.umd.edu; 3Sustainable Agricultural Systems Laboratory, USDA, ARS, Beltsville, MD 20705, USA; pat.millner@usda.gov; 4Department of Agriculture, Food and Resource Sciences, University of Maryland Eastern Shore, Princess Anne, MD 21853, USA; sparveen@umes.edu; 5Center for Food Safety and Security Systems, University of Maryland, College Park, MD 20742, USA

**Keywords:** soil microbiota, compost, soil biological amendments, agricultural farm, produce

## Abstract

As a traditional agricultural system, integrated crop-livestock farms (ICLFs) involve the production of animals and crops in a shared environment. The ICLFs in the mid-Atlantic region of the United States practice sustainable manure aging or composting processes to provide an on-farm source of soil amendment for use as natural fertilizer and soil conditioner for crop production. However, crop fertilization by soil incorporation of aged manure or compost may introduce different microbes and alter the soil microbial community. The aim of this study was to characterize the influence of aged or composted manure application on the diversity of soil bacterial community in ICLFs. Soil samples from six ICLFs in Maryland were collected before (pre-crop) and during the season (2020–2021) and used to analyze soil bacterial microbiome by 16S rDNA sequencing. Results showed that both phylum- and genus-level alterations of soil bacterial communities were associated with amendment of aged or composted manure. Particularly, Proteobacteria and Actinobacteria were enriched, while Acidobacteria, Bacteroidetes, Planctomycetes, Firmicutes, and Chloroflexi were reduced after manure product application. Meanwhile, the relative abundance of *Bacillus* was decreased, while two zoonotic pathogens, *Salmonella* and *Listeria*, were enriched by manure amendments. Overall, animal manure amendment of soil increased the phylogenetic diversity, but reduced the richness and evenness of the soil bacterial communities. Although manure composting management in ICLFs benefits agricultural sustainable production, the amendments altered the soil bacterial communities and were associated with the finding of two major zoonotic bacterial pathogens, which raises the possibility of their potential transfer to fresh horticultural produce crops that may be produced on the manured soils and then subsequently consumed without cooking.

## 1. Introduction

Integrated crop-livestock farms (ICLFs) are common types of commercial agricultural operations that are practiced worldwide in which both animal and crop production occur in proximity to each other on the same farm. Such farm operations are also referred to as mixed crop-livestock farming systems [1,2]. In recent years, the number of ICLFs have increased in the United States, particularly in the Northeast and Mid-Atlantic regions [3]. A majority of the ICLFs are either certified or non-certified organic, but in transition to organic farms, which largely contribute to the naturally pasture-based food supply chain in the nation, particularly for fresh produce and meat products including chicken, beef, and lamb [4]. Most of these food products are sold in farmers markets or local retail stores. According to USDA reports, more than 8000 farmers markets are currently listed in the National Farmers Market Directory [5,6].

Several researchers have reported that the cross-contamination levels of products from ICLFs and sold in farmers market with various zoonotic pathogens exceed those from conventionally grown products [7,8]. Such bacterial pathogen contamination in such food products may play an important role in sporadic or localized foodborne outbreaks due to the prohibited use of chemicals/antibiotics in certified organic production systems [9]. In addition, the close proximity of animal and produce operations in the same agricultural environments presents a potential for cross-contamination by zoonotic pathogens from farm animals to fresh produce [10].

The Centers for Disease Control and Prevention (CDC) reported more than 35 produce-relevant foodborne illness outbreaks in the US from 2006 to 2016 [11]. Fresh fruits and vegetables, including leafy greens, were responsible for more than half of the recorded foodborne outbreaks in the US [11]. Fresh fruits and vegetables which are typically consumed raw, are recognized as food products with the potential to become contaminated by several common foodborne pathogens such as *Salmonella*, shiga-toxin producing *Escherichia coli* (STEC), *Clostridium*, and *Listeria* [12] because of their exposure to pathogen sources in open-field production as well as during post-harvest handling practices. These enteric bacterial pathogens are part of the commensal microbiota of various farm animals, including poultry, cattle, pig, and goat, as well as wildlife. These pathogens generally can survive in partially or uncomposted animal waste and contaminate environments particularly soil and water, as well as fresh produce grown on contaminated soil, subjected to dust from such soil, and/or irrigated with contaminated water [13].

Most ICLFs develop and practice sustainable manure management, using manure-based soil amendments to fertilize soils to grow fresh produce crops [14,15]. Although fertile/cultivated soil is a rich source of microbes and microbial diversity, acting as the reservoir of microbial and genetic traits [16], most bacterial pathogens are not able to proliferate under such stress. However, it is noted that several opportunistic pathogens particularly *Salmonella*, STEC, *Listeria* and *Clostridium* in manure can persist in relatively low abundances but proliferate once under favorable conditions such as following the field application, though composting may properly reduce the majority of human pathogens below undetectable levels [17,18]. Although the animal manure containing compost serves as the fertilizer and improves soil health through enhancing the organic matter and accompanying properties, it plays critical roles in introducing various zoonotic pathogens, perpetuating pathogen reservoirs in livestock, extending their survival in the soil, and facilitating the microbial transfer from soil to fresh produce crops [13,19]. Specifically, using inadequately composted animal manure during recycling in ICLFs is associated with increased survival potential of pathogens in the manured soil [20]. This could be revealed by the higher contamination load of produce samples from organic integrated farms than that from organic produce-only farms containing no livestock in Europe [4]. Furthermore, the enteric pathogen contamination sources of fresh produce have been frequently traced back to environmental reservoirs with wild animals or agricultural farm operations [7,8,16]. Therefore, the structure and diversity of microbial community in soil prior and following manure/compost application need to be evaluated.

In this study, the soil bacterial microbiome in production fields on ICLFs was characterized before and after animal manure amendments were applied. Both pre-season and during-season soil samples were collected from ICLFs in the eastern and central Maryland areas, and the soil microbial composition and diversity were systematically compared. The comparison analysis of soil bacterial community profiles in ICLF produce fields can provide insight into potential conditions conducive to competitive exclusion and major population declines of targeted zoonotic pathogens that are the major risk factors for foodborne outbreaks.

## 2. Materials and Methods

### 2.1. Sample Collection and Processing

Soil samples were collected from six Maryland ICLFs, located in Accokeek, Clarksville, Pittsville, Princess Anne, Tyaskin, and Upper Marlboro, of which half were collected before and half after the application of manure products during the growing season of the plants. Multiple locations were picked at each farm and two separate visits in a week interval were made for biological replicates. The time interval of collection between before and after manure application varied farm to farm, but ranged from 90 to 120 days. Total 80 (collected from 40 locations, duplicate) soils were collected from multiple locations in cropping areas from each farm site on two separate visits. Not all these farms kept records of the temperatures achieved in their manure piles, or the total time manure was aged. One farm purchased spent mushroom compost produced commercially using a thermophilic process from horse manure for incorporation into their horticultural crop fields prior to planting/transplanting. Another site used their on-farm poultry and swine manure mixture, aged one year for their horticultural fields. Another site had poultry manure also aged one year for horticultural crop soil amendment prior to planting/transplanting. Manure product amendments were applied to planting row beds at rates that approximated slightly less than 5 T/ac to adhere to the phosphorus and nitrogen application recommendations. All the farms raised one or more agricultural animals, e.g., cattle, swine, chicken, turkey, and/or goat, on the farm, and they also grew a variety of horticultural products, ranging from leafy greens, herbs, tomatoes, peppers, melons, and corns.

On each farm, two replicate soil samples (approximately 1 kg) collected aseptically from adjacent surface sites, each approximately 12 cm diam × 8–10 cm deep, using sterile plastic scoops, were transferred into sterile Whirl Pack bags, and placed in a cooler containing frozen gel-packs for transport to the laboratory for further processing and analysis. Soil types ranged from sandy loam, sandy clay loam, to clay loam.

### 2.2. 16S rRNA Sequencing

Bacterial genomic DNA was extracted from each individual soil sample (300 mg) using the PureLink Microbiome DNA Purification Kit (Invitrogen, Carlsbad, CA, USA) following the manufacturer’s instructions. The replicate samples of extracted DNAs from the same soil site were combined and designated as one DNA sample. The amplification of bacterial gene-specific sequences was carried out based on the 16S rRNA variable V3 and V4 regions using 2× KAPA HiFi HotStart ReadyMix (KAPA Biosystems, Wilmington, MA, USA), for subsequent next-generation sequencing-based phylogenetic classification and diversity analysis. Amplification of the target V3-V4 16S rRNA region was conducted using primer pairs: TCGTCGGCAGCGTCAGATGTGTATAAGAGACAGCCTACGGGNGGCWGCAG and GTCTCGTGGGCTCGGAGATGTGTATAAGAGACAGGACTACHVGGGTATCTAATCC, running a program of 95 °C for 3 min, 25 cycles of 95 °C (30 s), 55 °C (30 s), and 72 °C (30 s), and finally 72 °C for 5 min. After cleaning up the amplicons with AMPure XP beads (Beckman Coulter Genomics, Danvers, MA, USA), dual indices and adapters were connected to each of the amplicons using Nextera XT Index Kit v2 Set C (Illumina, San Diego, CA, USA) by implementing a second PCR program of 95 °C for 3 min, 8 cycles of 95 °C (30 s), 55 °C (30 s), and 72 °C (30 s), and finally 72 °C for 5 min. Following the second clean-up using AMPure XP beads, the equimolar-pooled DNA libraries were prepared using Nextera XT DNA Library Preparation Kit (Illumina, San Diego, CA, USA). The constructed DNA library was mixed with PhiX Control v3 (Illumina, San Diego, CA, USA) as a reference and heated at 96 °C for 2 min for denaturation. The denatured DNA library was loaded into the cartridge for paired-end sequencing (2 × 300 bp) based on Illumina MiSeq system using v3 600-cycle kit (Illumina, San Diego, CA, USA). All the raw sequences were submitted to GenBank SRA under BioProject PRJNA731464, BioSample SAMN19285737 and SAMN19285738.

### 2.3. Metagenomic Dataset Processing

Dataset processing was performed in accordance with the method previously described [21,22]. The raw sequence dataset was demultiplexed using BCL2FastQ and the PhiX sequence was removed using DeconSeq. The separate paired FASTQ files were further filtered and trimmed using mothur (version 1.44) toolsuites. The problematic reads, including contigs longer than 250 bp and ambiguous bases, were removed using Screen.seqs tool for minimizing biases. The unique reads were trimmed using the Screen.seqs tool to ensure the overlapping of all reads with the 16S rRNA V3–V4 region. The gap characters and overhangs were removed using the Filter.seqs tool, and the near-identical sequences with under a threshold of 1% mismatches were merged using the Pre.cluster tool. The chimera hybrid sequences generated by mis-priming were then removed from the dataset using the VSEARCH algorithm. Taxonomic classification was conducted using the Classify.seqs tool, matching with SILVA reference database (version 138) based on the RDP classifier algorithm (version 11). Subsequently, sequence contaminations such as 16S/18S rRNA gene fragments from archaea, chloroplasts, and mitochondria were filtered using the Remove.lineage tool. A 97% identity threshold was applied for the Operational Taxonomic Units (OTUs), for clustering the 16S rRNA gene sequence variants.

### 2.4. Bacterial Taxonomy and Diversity Analyses

The relative abundance of a specific taxon was normalized and calculated as the Number of Reads for a taxon divided by the Number of Reads in Total 16S rRNA gene. The differences of bacterial abundances at both phylum and genus (the top 32 most abundant genera) levels between groups were determined by implementing the function ‘Analysis of Compositions of Microbiomes with Bias Correction (ANCOM-BC)’ using ANCOM package in R software [23]. The randomly sub-sampled taxonomic sequences (rarefaction depth = 13,800 sequences) were applied to calculate alpha and beta diversities. Various alpha diversity indexes including bergerparker, invsimpson, npshannon, and qstat measuring phylogenic diversity, ace, bootstrap, chao, and sobs assessing microbial richness, and heip, shannoneven, simpsoneven, and smithwilson evaluating microbial evenness, in soil bacterial microbiome were assessed through Summary. Single tool in mothur and compared between groups by one-way analysis of variance (ANOVA) using vegan package in R software [23]. Beta diversity was calculated based on the calculator the tayc/jclass and Phylip distance matrix using Dist.shared tool in mothur, followed by implementing analytical functions ‘non-metric multi-dimensional scaling (NMDS)’ and ‘analysis of similarities (ANOSIM)’ using vegan package in R software [24]. Multivariate analysis of variance based on permutations was conducted by implementing the function ‘Pairwise Permutational Multivariate Analysis of Variance (PERMANOVA)’ using CRAN package in R software [25]. Two-way Venn diagram was generated based on level 0.03 (97% similarity at species level) enclosing all datasets categorized in the two groups, implementing the function ‘VennDiagram’ using CRAN package in R software [26]. Statistically significant differences between groups were determined based on the *p* value less than 0.05.

## 3. Results

### 3.1. Relative Abundances of Soil Microbial Phyla

The relative abundances of all the bacterial phyla identified before and after the application of animal manure products only differed slightly (Figure 1). In general, the soil bacterial community was dominated by four major phyla, Proteobacteria, Acidobacteria, Actinobacteria, and Bacteroidetes. The relative abundances of these dominant phyla in soils were changed after the application of animal manure products (Figure 1). Specifically, the relative abundances of Proteobacteria, Acidobacteria, Actinobacteria, and Bacteroidetes in the soil samples collected before compost addition were 26.67%, 16.45%, 11.71%, and 8.36%, respectively, while those after addition of animal manure products were found to contain 29.57%, 15.13%, 14.67%, and 7.75%, respectively. Abundances of some other phyla including Chloroflexi, Firmicutes, Planctomycetes, and Verrucomicrobia were found to be decreased by 43.18%, 10.19%, 10.89%, and 12.35%, respectively, in association with manuring of the soil prior to planting (Figure 1).

### 3.2. Genera Compositions of Soil Microbiome

Slight differences in relative abundances of bacteria at the genus level were observed in the community analyses of the soil samples collected from pre-season (before adding the animal manure products) and during-season (after adding animal manure and during plant growth) from ICLFs. Overall, the genera displayed relatively even distribution in pre- and post-amendment soil samples. The top 32 most abundant genera with relative abundances in the soil samples collected from the pre- and post-amended groups were compared in this study (Figure 2A). Among the abundant bacterial genera, *Aquisphaera*, *Aridibacter*, *Brachybacterium*, *Brevibacterium*, *Burkholderia*, *Diplorickettsia*, *Faecalibacterium*, *Flavisolibacter*, *Gaiella*, *Gemmatimonas*, *Listeria*, *Nocardioides*, *Pedobacter*, *Rhodoplanes*, *Salmonella*, *Solirubrobacter*, *Spartobacteria*, *Sphingomonas*, *Streptomyces*, and *Terrimonas* were significantly (*p* < 0.05) increased in the soil samples post-amendment, by 35.33%, 40.97%, 66.47%, 273.35%, 61.67%, 880.92%, 27.49%, 98.31%, 65.35%, 16.99%, 157.14%, 52.07%, 33.91%, 18.15%, 833.33%, 25.44%, 22.61%, 19.66%, 45.22%, and 13.29%, respectively. In contrast, the relative abundances of *Arthrobacter*, *Bacillus*, *Bradyrhizobium*, *Nitrolancea*, *Parcubacteria*, *Povalibacter*, *Rhizomicrobium*, *Rhodopirellula*, and *Saccharibacteria* were significantly (*p* < 0.05) decreased in the soil samples post-amendment, by 27.92%, 39.62%, 23.34%, 49.14%, 38.03%, 17.17%, 15.47%, 58.73%, and 14.81%, respectively. The relative abundances of the remaining genera were irregularly distributed between the pre- and post-amendment groups. When the relative abundance of the top 32 predominant genera was compared among the six ICLFs, there were minor differences (Figure 2B). When comparing among the farms, in two farms (ICLF-1 and ICLF-5), the top abundant genera were present in relatively less percentage both before and after the application of the animal manure. A more similar pattern of distribution of the genera was observed in the ICLFs within each treatment group. The difference of genera among the farms in the same season might be attributed to the variety of plants grown including leafy greens, herbs, tomatoes, peppers, melons, and corns in different farms amid other reasons.

### 3.3. Species Phylogenetic Diversity, Richness, and Eveness in Soil Bacterial Communities

We identified various levels of differences and variations in the collected soil samples between pre- and post-amendment of animal manure products (Figure 3). Overall, the number of bacterial sequences obtained from pre-amendment samples was 38,958 ± 23,050 (88.01% coverage), in comparison with 35,081 ± 22,797 sequences (93.31% coverage) of the post-amendment samples. It was found that the phylogenetic diversity indexes in pre-amendment soil samples were generally smaller than those in post-amendment samples; the most significant difference was found in the npshannon index (*p* < 0.05), with 7.01 (post-amendment) vs. 5.76 (pre-amendment), while the remaining indexes indicated a more prominent phylogenetically diverse bacterial community in post-amendment samples than pre-amendment samples. The bacterial community richness reduction was reflected by the relatively smaller bootstrap and sobs indexes in post-amendment samples, in comparison with those in pre-amendment samples, in which the difference in sobs index was statistically significant (*p* < 0.01), whereas ace and chao indexes exhibited an opposite trend, showing higher values in post-amended samples, rather than in pre-amended samples. As for microbial evenness, no significant differences were identified between these two types of soil samples, while the soil bacterial microbiome in post-amended samples displayed an overall lower evenness than that in pre-amended samples; all the alpha diversity indexes indicating evenness were reduced in post-amended samples.

Additionally, the commonness of soil bacterial species between the two groups is displayed in Figure 4. The bacterial species were unorthodoxly distributed in all the soil samples, with a total of 8317 species identified. The soil samples from pre-amended ICLFs included 7062 bacterial species, while those from post-amended ICLFs included 7654 microbial species. Among these bacterial species, 6399 were shared between these two groups of soil samples, accounting for 76.94% of the ratio for commonness.

### 3.4. Dissimilarities in the Clustered Soil Microbial Compositions

The NMDS based on Bray–Curtis distances among grouped soil bacterial taxa is shown in Figure 5. The intra-group variation of dissimilarity for microbial composition among post-amended samples collected from ICLFs is higher than that among pre-amended samples. Furthermore, PERMANOVA indicated an insignificant (*p* = 0.305) difference in the composition of soil bacterial species between pre- and post-amended samples.

## 4. Discussion

Bacteria are the most abundant microorganisms in soil, while their community is largely influenced by soil properties, such as pH value, mineral content, temperature, and moisture [27,28]. Therefore, variations in soil microbiota are usually associated with different types of soils under differential environments and different plantations [29]. It is suggested that Proteobacteria and Acidobacteria are the most common and predominant phyla in soil bacterial communities [30]. A similar dominance was also observed in our study, where Proteobacteria dominated more than 25% of the entire soil bacterial composition, and Acidobacteria dominated more than 15% in the soil samples collected both before and after addition of animal manure-containing products. Such a similarity in the top two dominant soil bacterial phyla demonstrated the relative accuracy of methodology performed and the appropriation of pre-amended soil collection from ICLFs. Soil microorganisms, both directly and indirectly, play an important role in soil fertility/the food security, sustainability/soil health, quality and safety of food products as well as environment, which also may influence human health and nutrition [31]. However, the application of animal manure products as top-dressing during plant growth may, depending on the disinfection efficacy of any pre-treatment implemented prior to amendment, introduce shifts in the soil bacterial community that potentially increase the survival of pathogens. Thus, the composition and diversity of soil bacteria before and after the application of animal manure products into fresh produce horticultural farms, particularly ICLFs, is important to produce safe, heathy, nutritious fresh produce.

Our data indicated that the phylum-level abundances of soil microbes from ICLFs were noticeably altered by the application of animal manure products. This study specifically exhibited the increasing abundances of Actinobacteria and Proteobacteria, as well as the decreasing abundances of Planctomycetes, Chloroflexi, and Acidobacteria associated with addition of animal manure product amendments of horticultural soils. Actinobacteria are one of the dominant soil bacteria that facilitate elemental mineral cycling, especially carbon, nitrogen, phosphorus, and potassium [32]. Selected Actinobacteria have been previously employed for cattle, swine, and sheep manure composting as organic fertilizers promoting litter sanitation [14,32]. Proteobacteria, covering most of the zoonotic pathogens, are dominant in animal manures, especially cattle manure [33]. Planctomycetes have been widely spread from aquatic environments into soils, while they retain the functional role in slowly degrading various biopolymers [34]. Chloroflexi relies on photosynthesis and survives in soils with poor fertility, while their ecological function in soil is still unclear [35,36]. Acidobacteria in the plant–soil ecosystem actively modulate biogeochemical cycles (e.g., carbon, nitrogen, and sulfur cycles) [37]. Accordingly, based on phylum-level analysis, animal manure product application in ICLF soil tends to enrich Actinobacteria, which may accelerate soil nutrient cycling, but it can also introduce zoonotic pathogens, and lead to the loss of other functional soil bacteria and possibly other microbes.

In terms of soil bacterial genera, this study identified no genus with a relative abundance higher than 5% in pre-, post-amendment soil or in any individual farm regardless application of the manure, indicating an evenly taxonomic distribution of soil microbes in ICLFs. Several of the functional genera, such as *Gaiella*, *Brevibacterium*, *Diplorickettsia*, *Brachybacterium*, *Spartobacteria*, *Sphingomonas*, and *Nocardioides*, were substantially enriched by animal manure product application; these bacterial genera either contribute to nutrient cycling by decomposing complex biomass materials, or participate in nitrogen fixation that benefits plants or produce [38,39]. In contrast, the relative abundance of *Bacillus* was substantially diminished after the application of animal manure products in the soil; According to previous studies, *Bacillus* form spores and survive in soil to facilitate plant protection against biotic stresses and plant diseases caused by pathogens [40,41]. Reduced abundance of *Bacillus* associated with roots, or the rhizosphere may have negative impacts on plant production and pest defense [41]. Meanwhile, although the abundances of common zoonotic bacterial pathogens were low, we detected that the relative abundances of *Salmonella* and *Listeria* were enriched by 9.33 and 2.57 fold, respectively, following the application of manure products. This agrees with previous reports that implicate the use of manure products that are incompletely or inadequately treated by a validated ‘Process to Further Reduce Pathogens’ in the increased risk of *Salmonella* and *Listeria* contamination of fresh produce products typically consumed raw (i.e., without cooking) [20,42,43].

The biomass and diversity of soil microbial communities serve as a major driver of fundamental processes in soil, specifically nutrient cycling and decomposition of organic matter [44]. Previously, the application of cattle manure was shown to improve soil bacterial diversity and regulate the community structure in a tea plantation [45]. Application of swine manure was shown to enrich soil bacterial diversity by introducing manure-borne bacteria [46]. In this study, we observed a limited number of microbial sequences, while relatively high phylogenetic diversity but low richness and evenness in soil samples collected post-manure amendment. This indicates that the application of on-farm manure products in ICLFs could substitute for many indigenous soil bacteria, and that the manure amendment increased the phylogenetic relativeness among soil microbes, disrupting the original rich contents as well as the overall even distribution of soil bacteria.

## 5. Conclusions

Enriched abundances of Proteobacteria and Actinobacteria, together with reduced abundances of Acidobacteria, Bacteroidetes, Planctomycetes, Firmicutes, and Chloroflexi in the soil microbiome at the phylum level, are associated with the application of manure products in ICLFs. Significant genus-level shifts occurred from the use of manure product amendments in soil. However, the potential risk of introducing foodborne illness pathogens such as *Salmonella* and *Listeria* remains. Overall, the application of animal manure to the ICLFs expands divergent lineages and compromises the original richness and even distribution of soil bacteria. Meanwhile, the findings from this study provide farmers and risks assessors with insights on the effects of manure product amendments on soil bacterial community changes in small-scale private commercial ICLFs. This evaluation sets the stage for further in-depth investigations of the relationships among the altered soil bacterial communities resulting from manure amendment and key soil microbial functions, as related to nutrient transformations, soil carbon cycling, and quality and safety of produce from ICLFs.

## Figures and Tables

**Figure 1 microorganisms-09-02586-f001:**
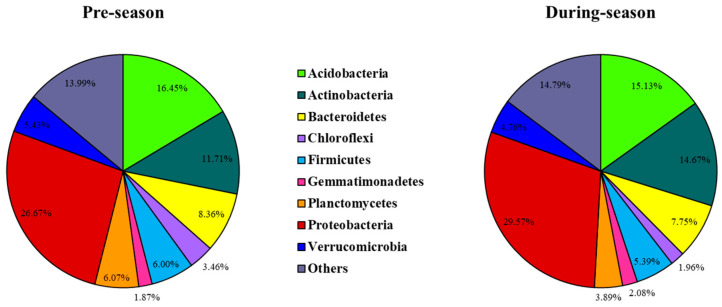
Relative percentages of the phylum level abundances in soil microbiome before (**left**) and after (**right**) the application of animal manure product.

**Figure 2 microorganisms-09-02586-f002:**
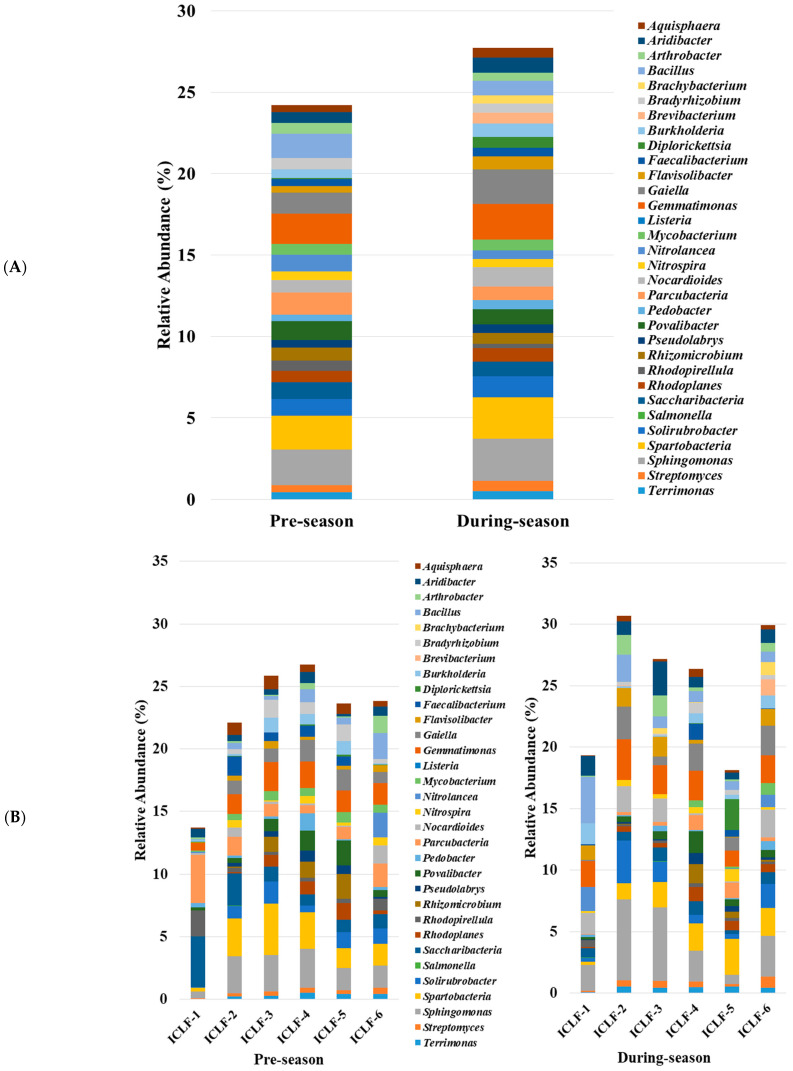
Relative abundances of the top-abundant soil bacterial genera before (in the left) and after (in the right) the application of animal manure product (**A**) combining all the six farms; (**B**) in different farms (ICLF 1 to ICLF6 represents individual farm).

**Figure 3 microorganisms-09-02586-f003:**
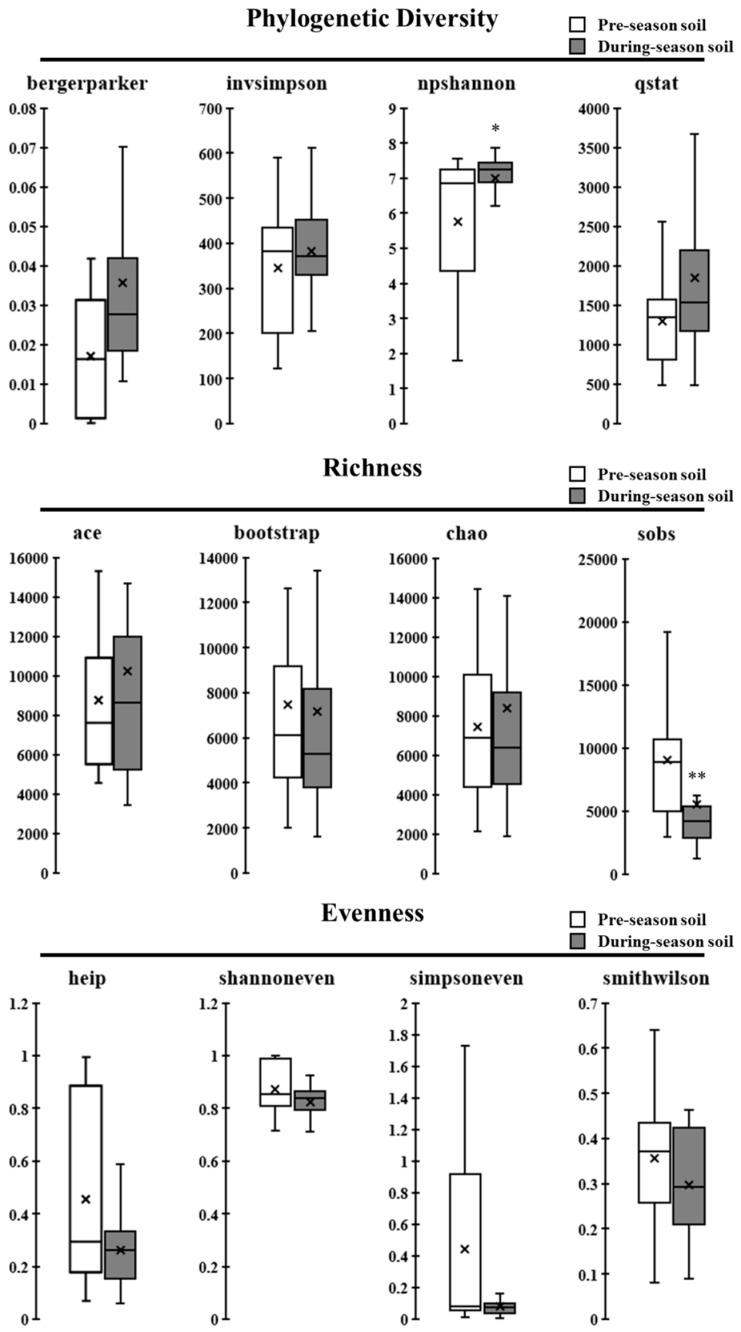
Bacterial diversity in the pre- and post-amended soil communities. Alpha diversity indexes of bergerparker, invsimpson, npshannon, and qstat (phylogenic diversity), ace, bootstrap, chao, and sobs (richness), and heip, shannoneven, simpsoneven, and smithwilson (evenness) for pre- and post-amended soil samples. * and ** indicate significant differences in comparison with pre-amended soil at *p* < 0.05 and *p* < 0.01, respectively. × indicates the mean value of the group.

**Figure 4 microorganisms-09-02586-f004:**
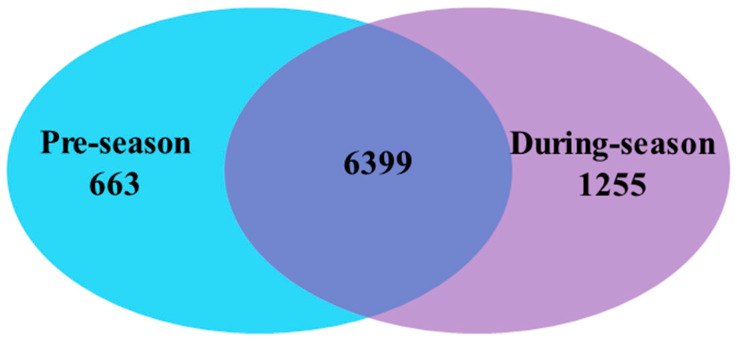
Two-way Venn diagram for bacterial commonness of soil bacterial species between pre-season and during-season soil samples.

**Figure 5 microorganisms-09-02586-f005:**
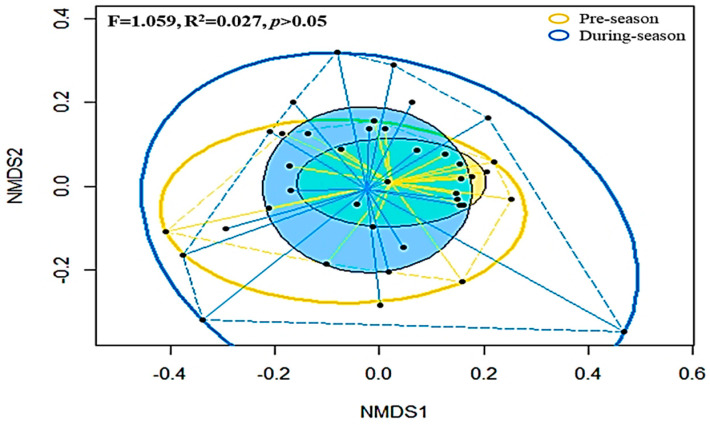
Non-metric multidimensional scaling (NMDS) based on Bray–Cutis distance matrix enclosing all 40 datasets. Black dots represent individual soil bacterial community; Solid lines represent ellipses enclosing all points in different groups; Dashed lines represent convex hulls by ordihull function.

## Data Availability

The data presented in this study are openly available in GenBank SRA under BioProject PRJNA731464, BioSample SAMN19285737 and SAMN19285738.

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
