# Peer review of "Influence of Manure Application on the Soil Bacterial Microbiome in Integrated Crop-Livestock Farms in Maryland"

_microorganisms, 2021, doi:10.3390/microorganisms9122586_

Round 1

Reviewer 1 Report

I appreciate the effort made by the authors. Based on the design performed, the data and results generated can be considered correct. However, I suggest that for future works, information from samples of different nature should not be grouped in a single data and that determining factors, in addition to the one under study, should be taken into consideration.

Author Response

We thank the reviewer for their approving comment. We really appreciate the reviewer for their valuable insight and we’ll keep the suggestion in mind for future project design.

Reviewer 2 Report

Dear Authors,

I have some comments and suggestions to your manuscript, following:

  • Line from 196 to 203: The differences between the numbers of different groups of bacteria in the experimental variants are so small, but are they statistically significant?
  • Please supplemented section -2.1. Sample collection and processing.

Describe in detail the sampling time, how many days have elapsed between sampling , before and after manure application. Additionally, did you take samples after applying manure during the growing season of the plant?

  • Line 210: What plants grew during -season?
  • The microbial diversity in the soil after manure application during the plant growth period did not depend only on manure but also growing plant/.
  • Line 287: ..soil properties (or soil features)… but not soil composition,

not only, the abundance of microorganisms also depends on the type of crop.

Reviewer.

Author Response

We thank the reviewer for their valuable comments and recommendations. According to the recommendations, we have changed the text and marked in blue color front. Below is the point by point answer to the reviewer’s comments.

  • Line from 196 to 203: The differences between the numbers of different groups of bacteria in the experimental variants are so small, but are they statistically significant?

Answer: The differences between the numbers of different groups of bacteria in the experimental variants are small and differ only numerically. The differences were not statistically significant for line 196 to 203.

  • Please supplemented section -2.1. Sample collection and processing. Describe in detail the sampling time, how many days have elapsed between sampling , before and after manure application. Additionally, did you take samples after applying manure during the growing season of the plant?

Answer: In the same season (either before or after application of the manure), sampling were done within 1 week interval in duplicate visits. For time interval of sample collection between before and after manure application varied farm to farm but within 90 to 120 days. Yes, we collected the sample after applying manure during the growing season.

All the above mentioned information is now added in the method materials section of the text in blue color front.

  • Line 210: What plants grew during -season?                                             The microbial diversity in the soil after manure application during the plant growth period did not depend only on manure but also growing plant.

Answer: Different plants including leafy greens, herbs, tomatoes, peppers, melons, and corns were grown in different farms. Yes, this could be one of the reasons of differences in genera among the farms in the same season and now the reason is included in the result section in blue color front.

  • Line 287: ..soil properties (or soil features)… but not soil composition,

not only, the abundance of microorganisms also depends on the type of crop.

Answer: Thank you for the suggestion. We have changed the ‘soil composition’ with ‘soil properties’ and included type of crop as a reason of difference. All changes are marked in blue color front.

This manuscript is a resubmission of an earlier submission. The following is a list of the peer review reports and author responses from that submission.

Round 1

Reviewer 1 Report

The starting point of this study, although not especially relevant from a scientific perspective, shows some interest from a more applied point of view. Translation of results to organic farmers in order to optimize their practices and resources is a good point. However, the experimental design and the analysis format do not allow to draw any firm conclusion on this respect. There are variables that authors have not considered which probably influence the composition of microbial communities, such type of soil, manure, the origin and state of manure, etc.

On the other hand, the depth of analysis clearly shows room for improvement. Discussion is a little bit poor.

Grammar corrections:

  • The use of puntuaction marks is quite defficient.
  • 22-23: “… with by amend-22 ment with aged or composted manure”.
  • 59: “… which are typically are…”.
  • 61: “… by several by common foodborne…”
  • 67: “…fresh produce which is…”.
  • 199-205: “Specifically, the relative abundances of Proteobacteria, Acidobacteria, Actinobacteria, and Bacteroidetes in the soil samples collected before compost addition were 26.67%, 16.45%, 201 11.71%, and 8.36%, respectively. Whereas the relative abundances of Proteobacteria, Acidobacteria, Actinobacteria, and Bacteroidetes in the soil samples collected after addition of animal manure products (during crop growing periods), were found to contain 29.57%, 15.13%, 14.67%, and 7.75%, respectively”. It´s not necessary to repeat bacterial phyla, as well as make clear all the time that pre- and post-manure application correspond to pre-growing and durin growing is also repetitive.

General suggestions:

  • Microflora is not the appropiate term to refer to microorganisms. Please, use microbiota.
  • Scientific names should be in Italics.
  • Section 2.1. is repeated.
  • Symbol of degree Celsius is wrong.
  • Include primers used at amplification processes.
  • There are a complete absence of bibliographic references at Materials and Methods.
  • Decrease ratio calculations for different microbial taxa are not correctly expressed. For example, relative abundances for Chloroflexi phylum are 3.46% and 1.96%, respectively, which according to authors means a decrease of 1.49%. If they express difference as a percentage, the correct value is 56.94%.
  • In the text, it is said that difference between values for sobs index is significant, but in Figure 3 it´s not marked with an asterisk.

Author Response

Reviewer 1

The starting point of this study, although not especially relevant from a scientific perspective, shows some interest from a more applied point of view. Translation of results to organic farmers in order to optimize their practices and resources is a good point. However, the experimental design and the analysis format do not allow to draw any firm conclusion on this respect. There are variables that authors have not considered which probably influence the composition of microbial communities, such type of soil, manure, the origin and state of manure, etc. On the other hand, the depth of analysis clearly shows room for improvement. Discussion is a little bit poor.

Response: We thank the reviewer for both positive and negative feedback on our study. In response to the reviewer’s comments, we have thoroughly revised and improved our manuscript. However, due to the limitation of time and feasibility, we could not change our experimental design or involve more variables for investigation. A further in-depth analysis involving multiple variables of soil and manure compost will be our long-term target for future studies.

Grammar corrections:

The use of puntuaction marks is quite defficient.

Response: We have modified all the punctuation throughout the manuscript.

22-23: “… with by amend-22 ment with aged or composted manure”.

Response: We have corrected it at Line 23.

59: “… which are typically are…”.

Response: We have corrected it at Line 60.

61: “… by several by common foodborne…”

Response: It has been corrected at Line 61.

67: “…fresh produce which is…”.

Response: It has been corrected at Line 67.

199-205: “Specifically, the relative abundances of Proteobacteria, Acidobacteria, Actinobacteria, and Bacteroidetes in the soil samples collected before compost addition were 26.67%, 16.45%, 201 11.71%, and 8.36%, respectively. Whereas the relative abundances of Proteobacteria, Acidobacteria, Actinobacteria, and Bacteroidetes in the soil samples collected after addition of animal manure products (during crop growing periods), were found to contain 29.57%, 15.13%, 14.67%, and 7.75%, respectively”. It´s not necessary to repeat bacterial phyla, as well as make clear all the time that pre- and post-manure application correspond to pre-growing and durin growing is also repetitive.

Response: We thank the reviewer for pointing out this issue. In accordance with the comments and suggestions, we have thoroughly revised the description of results at Lines 200-207.

General suggestions:

Microflora is not the appropiate term to refer to microorganisms. Please, use microbiota.

Response: Thank you for the suggestion. We have changed ‘microflora’ into ‘microbiota’ in the text.

Scientific names should be in Italics.

Response: Thank you. We have checked throughout the manuscript and italicized all the scientific names.

Section 2.1. is repeated.

Response: Sorry about this mistake. The repetitive content has been deleted.

Symbol of degree Celsius is wrong.

Response: We have corrected all the symbols of Celsius degree.

Include primers used at amplification processes.

Response: Thank you for pointing this out. We have included the primer sequences for PCR amplification at Line 134-137.

There are a complete absence of bibliographic references at Materials and Methods.

Response: Thank you for providing this comment. Accordantly, we have cited relevant literature in the Materials and Methods section.

Decrease ratio calculations for different microbial taxa are not correctly expressed. For example, relative abundances for Chloroflexi phylum are 3.46% and 1.96%, respectively, which according to authors means a decrease of 1.49%. If they express difference as a percentage, the correct value is 56.94%.

Response: Thank you for this constructive comment. We have corrected the expression of differences in percentage for both increasing and decreasing values.

In the text, it is said that difference between values for sobs index is significant, but in Figure 3 it´s not marked with an asterisk.

Response: Thank you for pointing out this mistake. We have marked and indicated the significant symbol in Figure 3.

Reviewer 2 Report

The manuscript presents the results of an assessment of the shifts in microbial communities in agricultural fields before and after manure application. I believe that the work is interesting and well-done, and in scope with the aims of the journal. I only have some minor comments that I put directly in the attached manuscript.

Author Response

Reviewer 2

The manuscript presents the results of an assessment of the shifts in microbial communities in agricultural fields before and after manure application. I believe that the work is interesting and well-done, and in scope with the aims of the journal. I only have some minor comments that I put directly in the attached manuscript.

Response: We thank the reviewer for recognizing the value of our study and providing constructive comments for our manuscript. In accordance with the comments, we have revised our manuscript and marked all the revisions in red color font.

Reviewer 3 Report

Dear Authors,

I would like to give some comments about your manuscript, the following:

Line 18: Suplement this sentense- The aim of this study was to characterize the influence of aged or composted manure  application on the soil bacterial community diversity in ICLFs.

Line  88 : „microbial diversity „ not „microbial ecosystems”…

”Therefore, the microbial diversity in soil prior and following manure/compost application need to be evaluated.

Lines 122 to 139 should be deleted, they are a repetition of „chapter 2.1”

Line 272: Delete the text: "* indicates significant differences in comparison with pre-

amended soil " from the legend in Fig. 3, or mark "*" in Fig. 3.

Although you talk about Salmonella sp. and Listeria sp. in Abstract, Introduction, Discussion and Conclusion, I cannot find the results for these bacteria in the Results section. Please complete the Results section.

Author Response

Reviewer 3

I would like to give some comments about your manuscript, the following:

Line 18: Suplement this sentense- The aim of this study was to characterize the influence of aged or composted manure application on the soil bacterial community diversity in ICLFs.

Response: Following your suggestion, we have modified the sentence at Line 19.

Line 88: „microbial diversity „ not „microbial ecosystems”…

”Therefore, the microbial diversity in soil prior and following manure/compost application need to be evaluated.

Response: In accordance with your comment, we have revised it at Line 90.

Lines 122 to 139 should be deleted, they are a repetition of „chapter 2.1”

Response: Thank you for pointing out this mistake. We have deleted the repetitive content.

Line 272: Delete the text: "* indicates significant differences in comparison with pre-amended soil "from the legend in Fig. 3, or mark "*" in Fig. 3.

Response: Thank you for pointing out this mistake. We have marked and indicated the significant symbol in Figure 3.

Although you talk about Salmonella sp. and Listeria sp. in Abstract, Introduction, Discussion and Conclusion, I cannot find the results for these bacteria in the Results section. Please complete the Results section.

Response: The result description including the altered relative abundances of Salmonella and Listeria genera has been revised and improved at Line 218-229.

Round 2

Reviewer 1 Report

Although the authors have implemented some of the suggestions made, especially regarding Materials ans Methods section, the basic objections are still present. The analysis aims to elucidate the effect caused by amedments of compost on the structure of the microbial community. To achieve this objective, it would be neccessary that the only variable be the incorporation or not of compost. However, several variables are present in this study, which may influence the results. Thus, conclusions attributing all the results to the presence of compost leave out other factors that may be equally important.

On the other side, paragraphs in the discussion marked as modified remain exactly the same (two last paragraphs).